# Desert Beetle-Inspired Hybrid Wettability Surfaces for Fog Collection Fabricated by 3D Printing and Atmospheric Pressure Plasma Treatment

**DOI:** 10.3390/biomimetics10030143

**Published:** 2025-02-26

**Authors:** Chia-Yi Lin, Ting-An Teng, Haw-Kai Chang, Po-Yu Chen

**Affiliations:** 1Department of Materials Science and Engineering, National Tsing Hua University, Hsinchu 300044, Taiwan; rainyi0109@gmail.com (C.-Y.L.); anniebobo17@gmail.com (T.-A.T.); hkai1616@gmail.com (H.-K.C.); 2Instrumentation Center, National Tsing Hua University, Hsinchu 300044, Taiwan

**Keywords:** fog collection, bio-inspired, Namib Desert beetle, cacti, 3D printing, atmospheric pressure plasma, hybrid wettability surface

## Abstract

Freshwater resources that humans can use directly account for 2.5 percent. Fog collection from the atmosphere is an eco-friendly and potential solution to the water shortage crisis. This study presents a biomimetic approach to fog collection inspired by the Namib Desert beetle and cacti. Using fused deposition modeling (FDM) 3D printing and atmospheric pressure plasma (APP) treatment, we fabricated hybrid wettability surfaces combining hydrophobic polypropylene (PP) and super hydrophilic polycarbonate (PC). These surfaces significantly improved fog collection efficiency, achieving 366.2 g/m^2^/h rates by leveraging the Laplace pressure gradient and hybrid wettability gradient. This work provides an efficient and effective methodology to fabricate hybrid wetting surfaces and can be potentially applied to fog harvesting and microfluidic devices.

## 1. Introduction

The scarcity of freshwater is becoming a severe global crisis with extreme weather, a growing population, and industrial development. More than 50% of the world’s population encounters this problem [1,2]. Fog harvesting or capturing water from the atmosphere is a potential solution to the shortage of freshwater. Therefore, more and more groups have been researching fog collection in recent years [3,4,5,6,7,8,9,10,11,12,13,14]. In nature, many creatures can collect water from the air without additional energy consumption. Researchers have studied their surface chemical compositions and characteristic surface morphology. They investigated bioinspired fog collection systems by mimicking animals and plants such as the Namib Desert beetles [3,4,5,10], cacti [7,9,12] and the Namib Desert grasses [13,14]. Namib Desert beetles have specific elytra composed of hydrophilic bumps and waxy hydrophobic surroundings. Fog droplets coalesce and grow on hydrophilic bumps, transported to the mouth through hydrophobic surroundings. Cacti use their hierarchical cone-like spines to move water droplets from the tip to the base, driven by the Laplace pressure gradient, the main driving force of conical and wedge-shaped structures [7,15]. Parallel grooves are also regarded as the transport channels for water droplets [13]. Similarly, Namib Desert grass utilizes hydrophilic parallel grooves on its leaves for water collection. Figure 1 shows the summary of the fog-harvesting capabilities of these creatures.

Surface patterns combining hydrophilic and hydrophobic regions have been widely studied for their role in enhancing fog collection efficiency. Previous studies have demonstrated that two-dimensional (2D) hybrid surfaces can significantly improve water collection through directional droplet transport [16,17,18]. Garrod et al. fabricated superhydrophilic-hydrophobic patterned surfaces mimicking the Namib Desert beetle’s back, demonstrating enhanced microcondensation efficiency [16]. Bai et al. developed integrative bioinspired surfaces with star-shaped wettability patterns, achieving superior water collection by combining surface energy gradients with Laplace pressure-driven transport [17]. More recently, Nioras et al. systematically examined micro-nanotextured hybrid surfaces, showing that the shape and arrangement of hydrophilic-hydrophobic patterns influence collection efficiency in both fog and dew harvesting modes [18].

Building upon these findings, this study aims to develop an economical and efficient method to fabricate patterned surfaces with hybrid wettability inspired by natural creatures. We utilized polypropylene (PP) and polycarbonate (PC) as materials to create bumpy structures inspired by the Namib Desert beetle through 3D printing of fused deposition modeling (FDM). Subsequently, atmospheric pressure plasma (APP) treatment was applied to the composite samples. The treatment preserved the polypropylene (PP) hydrophobicity while making the polycarbonate (PC) super hydrophilic, achieving hybrid wettability surfaces. We designed additional star-shaped structures and leaf-shaped structures inspired by the Namib Desert grass to improve the performance of fog collection. We applied the same process to synthesize hybrid wettability surfaces with different 3D structures. Laser confocal scanning microscopy was used to analyze the surface profiles of the bioinspired patterned structures, and the static water contact angles were measured as well. The chemical composition and bonding characteristics were analyzed by electron spectroscopy, and dynamic fog collection processes were recorded using a digital camera. This rapid and economical method can potentially be used in the fields of fog harvesting and microfluidic devices [19,20,21].

## 2. Materials and Methods

### 2.1. Preparation of Designed Pattern

In this study, all samples were printed on an FDM 3D printer, Ultimaker S3 (Ultimaker, Zaltbommel, The Netherlands), equipped with dual jets to print polypropylene (PP) and polycarbonate (PC) samples. The printing layer resolution was 0.1 mm, and the resolution in the X, Y, and Z directions was 6.9, 6.9, and 2.5 microns, respectively. The building range was 230 mm long, 190 mm wide, and 200 mm high. We used SolidWorks software (Dassault Systèmes S.A., Waltham, MA, USA) to design patterned structures, which were then exported as stereolithographic files (.stl) into the Ultimaker Cura 3D printer software (Ultimaker, Zaltbommel, The Netherlands). The printing process was described as follows: PP and PC filaments were first loaded into the 3D printer. The substrate and surface patterns materials were determined using computational software. The .stl files were cut with the 3D printer software for printing. The printer base plate was then heated to 100 °C so the samples could adhere tightly to the glass plate. PP and PC filaments were heated to melt and extruded by dual-jet printing. These extruded filaments were packed layer by layer as the designed files. The length and width of all samples are 32 mm, and the patterns were in a 5 × 5 array.

### 2.2. Surface Modification

The surface wettability of the printed samples was modified using an atmospheric pressure plasma (APP) system (SAP-013, Creating Nano Technologies, Taiwan). Clean dry air (CDA) was used as the working gas and was dehumidified using a freeze dryer. A plasma jet with 2–3 mm diameter and a length of about 1 cm was generated and blown out by the gas flow. The plasma jet scanned the surface at a velocity of 75 mm/s with a 3 mm interval, the power output was set to 350 watts, and the gas flow rate was 24 L per minute. The samples were treated at a working distance of 2 mm, resulting in PP maintaining hydrophobicity and PC becoming superhydrophilic.

### 2.3. Characterization

Several advanced characterization techniques evaluated the quality of printed samples and surface modifications. Optical Emission Spectroscopy (OES) was used to analyze the plasma composition by capturing optical signals of excited atoms, ions, and molecules, with the signals processed using OceanView software on an optical spectrometer (USB 2000+, Ocean Insight, Orlando, FL, USA). The surface topography of the 3D-printed samples was examined with a Keyence VK-X laser scanning confocal microscope (Keyence, Osaka, Japan), which delivers high-resolution, large depth-of-field color images with nanometer-level height precision, enabling accurate measurements of profiles, sizes, and roughness. Fourier-Transform Infrared Spectroscopy (FTIR) was utilized for surface qualitative analysis using a Jasco FTIR-4600 (Jasco, Tokyo, Japan) spectrometer to measure functional groups on the surface of composite samples after atmospheric pressure plasma treatment, covering a wavenumber range of 7800–$350\;{\mathrm{cm}}^{-1}$. The surface chemical composition of untreated PP, untreated PC, APP-treated PP, and APP-treated PC was characterized by electron spectroscopy for chemical analysis (ESCA) (PHI 5000 Versaprobe II, ULVAC-PHI). The wettability of the surfaces was determined using a goniometer (First Ten Angstroms, Newark, CA, USA) equipped with a tilting stage and CCD camera, with deionized water droplets (2μL) deposited on the surfaces at room temperature (24–26 °C) and relative humidity (50–65%), capturing images within five seconds after deposition.

### 2.4. Experimental Setup

A semi-enclosed chamber was set up for fog collection experiments, using a commercial humidifier (SC-EB35B, YADU, Beijing, China) to generate fog. The experiments were conducted at a room temperature of 25–26 °C and a relative humidity of 75–85%, with a fixed 10 cm distance between the fog outlet and the sample. The fog had a wind velocity of 0.3 m/s and a flow rate of 136 mL/h. An electronic balance (TX223L, Shimadzu, Kyoto, Japan) measured the mass of water collected over two hours, with three measurements per sample. The fog collection processes were recorded using a digital camera (D750, Nikon, Tokyo, Japan). To further visualize the process, colored DI water, mixed with methylene blue hydrate powder (Sigma-Aldrich^®^, St. Louis, MA, USA), was sprayed onto the samples with a sprayer.

## 3. Results

### 3.1. Surface Characterization

#### 3.1.1. Static Contact Angle Measurement

Six available materials could be printed together with the FDM 3D printer, namely polylactic acid (PLA), thermoplastic polyurethanes (TPU), polyethylene terephthalate carbon fiber (PETCF), polypropylene (PP), polycarbonate (PC), and nylon.

Figure 2 shows the water contact angles of these six materials before and within one minute after AP plasma treatment. After AP plasma treatment, the largest contact angle contrast appeared between the PP and PC samples. The intrinsic contact angles of PP and PC were 91.6° and 80.1°, respectively. PP maintained its hydrophobicity with the contact angle of 93.3° after AP plasma treatment, while PC became super hydrophilic with the contact angle of 3.2°. Consequently, PP and PC were chosen for the following processes. However, due to surface aging, the contact angle of the PC treated with APP increased to 18.4° after one day and gradually reached around 25° over the next 10 days. Afterward, it remained stable at 25°, maintaining its hydrophilicity rather than returning to the intrinsic contact angle of untreated PC (Appendix A).

#### 3.1.2. Fourier-Transform Infrared Spectroscopy

Figure 3 presents the FTIR spectra of untreated and APP-treated polypropylene (PP) and polycarbonate (PC). In Figure 3a, the spectra for untreated and APP-treated PP are nearly identical, showing characteristic peaks for C-H bending (769–839 cm^−1^), C-C stretching (974–997 cm^−1^), and CH_3_ stretching (2839–2873 cm^−1^), among others. Minimal changes were observed after plasma treatment, with slight decreases in the intensities of CH_3_ and CH_2_ stretching peaks. This suggests that PP exhibited limited reactivity to AP plasma, which explains why its surface maintained its original hydrophobicity post-treatment.

In contrast, Figure 3b highlights significant changes in the spectra of PC after plasma treatment, with notable increases in the intensities of peaks related to C-OH (1157 cm^−1^), C-O-C (1221 cm^−1^), and C=O stretching (1770 cm^−1^). These shifts indicate the formation of hydroxyl and oxygen-containing functional groups, which contributed to the activation of the PC surface, rendering it super hydrophilic. This transformation supports increased fog collection efficiency for plasma-treated PC surfaces.

#### 3.1.3. Electron Spectroscopy for Chemical Analysis

Figure 4 and Figure 5 show ESCA survey scans of untreated PP, APP-treated PP, untreated PC, and APP-treated PC. The C1s spectra for untreated (Figure 4a) and APP-treated PP (Figure 4b) showed peaks at 284.0 and 284.5 eV for C-C/C-H bonds in untreated PP, while APP-treated PP exhibited additional peaks at 283.6 and 286.6 eV, indicating C-O bonds and minor oxygen incorporation. Figure 4d shows the O1s spectra that revealed a low-intensity peak at 532.1 eV, corresponding to the hydroxyl groups generated during plasma treatment.

For PC, the C1s spectra included peaks for untreated PC (Figure 5a) at 283.6 eV (C=C), 284.0 and 284.4 eV (C-C/C-H), 285.4 eV (C-O-C), and 289.7 eV (carboxylic groups). APP treatment (Figure 5b) introduced new peaks at 286.6 eV (C-O) and 288.3 eV (C=O), indicating increased oxygen bonding and hydroxyl group formation. The O1s spectra shown in Figure 5c,d confirmed a higher intensity of O-C bonds compared to O=C bonds, consistent with oxygen radical generation during plasma treatment. These modifications transformed APP-treated PC into a superhydrophilic material by enhancing its ability to bind with water molecules.

### 3.2. Fog Collection Efficiency

#### 3.2.1. Namib Desert Beetle-Inspired Bumpy Structures

The bumpy structures were designed with a diameter of 4 mm, a center-to-center spacing of 6 mm, and a height of 2 mm (Figure 6a). Laser confocal images (Figure 6b) confirmed high printing accuracy, with the diameter, spacing, and height achieving 98.5%, 97.5%, and 96.7% precision, respectively. These results validate the effectiveness of the 3D printing process in fabricating the designed structures.

Figure 7a illustrates the fog collection performance of bumpy structures. Hybrid wettability samples, where PC forms the bumps and PP serves as the substrate (PC/PP), and vice versa (PP/PC), began collecting water within 30 min, outperforming homogeneous wettability samples (PP and PC). Superhydrophilic PC samples initially showed rapid collection but plateaued after 60 min, whereas hybrid samples maintained higher collection rates throughout the experiment. In Figure 7b, the water collection rate was determined at the end of the 120-min fog collection experiment. Among the tested surfaces, PC/PP exhibited the highest fog collection rate of 364.5 g/m^2^h, followed by PP/PC (347.7 g/m^2^h), PC (264.2 g/m^2^h), and PP (215.4 g/m^2^h).

The collection processes differed between the samples as shown in Figure 8, illustrating the fog collection mechanisms of four bumpy surface configurations with different wettabilities. In hydrophobic PP, water droplets nucleate, grow, and coalesce but remain pinned until they reach a critical size, leading to inefficient detachment. In contrast, superhydrophilic PC promotes spontaneous droplet spreading, forming a thin water film that drains slowly, limiting water removal efficiency. For superhydrophilic PC/hydrophobic PP, water droplets transfer from the PC bumps to the PP substrate, where droplets accumulate at the interface, coalesce, and eventually detach. The hydrophobic PP/superhydrophilic PC structure demonstrates the most efficient collection process: water accumulates on the hydrophilic substrate while droplets on hydrophobic bumps roll off and merge with the thin water film, facilitating continuous and efficient transport because of the wettability gradient. The hybrid structure outperforms homogeneous surfaces by leveraging hydrophobic regions for droplet nucleation and hydrophilic channels for rapid water transport, overcoming the limitations of slow drainage in superhydrophilic surfaces and poor detachment in hydrophobic ones.

#### 3.2.2. Three-Pointed Star-Shaped Structures

Three-pointed star-shaped structures were designed with a center-to-center spacing of 6 mm, a height of 2 mm, and an internal angle of 36°. To achieve comparable surface area to bumpy structures and to introduce a Laplace pressure gradient, the top and bottom lengths of the stars were set at 1.26 mm and 2.43 mm, respectively, in Figure 9a. Figure 9b shows the Laser confocal imaging of high dimensional accuracy for spacing (99.0%) and height (97.8%), though contour precision was lower (83.5%) due to limitations in the printing system, particularly at sharp tips.

Fog collection performance for three-pointed star-shaped structures is summarized in Figure 9. Hybrid wettability samples (PC/PP and PP/PC) began collecting water within 15 min, outperforming homogeneous wettability samples (PP and PC), which started at 30 min, as shown in Figure 10a. Among the samples, PP/PC showed the highest collection rate at 341.6 g/m^2^h, followed by PC/PP (298.6 g/m^2^h), PC (238.8 g/m^2^h), and PP (207.3 g/m^2^h) in Figure 10b.

Droplet behavior varied across samples, as shown in Figure 11. For hydrophobic PP, droplets coalesced randomly and accumulated at star centers due to the Laplace pressure gradient. However, many remain pinned until they reach a critical size, leading to inefficient transport. Superhydrophilic PC surfaces formed water films that accumulated at the bottom, dripping in elongated shapes, limiting collection efficiency. Hybrid samples (PC/PP and PP/PC) exhibited enhanced coalescence and transport, leveraging Laplace pressure and hybrid wettability gradients. However, the lack of a strong directional transport mechanism reduces the efficiency of PC/PP samples. PP/PC samples directed droplets from hydrophobic stars to superhydrophilic substrates, forming larger droplets and accelerating transport via water films.

#### 3.2.3. Five-Pointed Star-Shaped Structures

Five-pointed star-shaped structures were designed with a center-to-center spacing of 6 mm, a height of 2 mm, and an internal angle of 36°, as shown in Figure 12a. The top and bottom lengths of the stars were set at 1.03 mm and 1.94 mm, respectively, to maintain comparable surface area to bumpy structures and introduce a Laplace pressure gradient. Figure 12b shows the laser confocal images with high precision in spacing (98.6%) and height (98.4%), though contour accuracy was limited to 80.4% due to challenges in printing sharp tips.

Fog collection performance for five-pointed star-shaped structures is illustrated in Figure 12. In Figure 13a, Hybrid wettability samples (PC/PP and PP/PC) began collecting water within 15 min, while homogeneous wettability samples (PP and PC) started at 30 min or later. The water collection rate was shown in Figure 13b; PP/PC achieved the highest collection rate of 366.2 g/m^2^h, followed by PC/PP (304.7 g/m^2^h), PC (246.3 g/m^2^h), and PP (220.2 g/m^2^h).

Droplet behaviors for five-pointed star-shaped samples are shown in Figure 14. In hydrophobic PP, droplets coalesce, grow, and accumulate at the star center due to the Laplace pressure gradient, with most forming in concave regions before coalescing and rolling off. The superhydrophilic PC forms a thin water film that drains slowly, limiting collection efficiency. In superhydrophilic PC/hydrophobic PP, water flows from hydrophilic stars to the hydrophobic substrate, where droplets gather, coalesce, and detach, although transport remains inefficient. The PP/PC hybrid structure achieves the best performance by using the Laplace and dual wettability gradient to guide droplets from hydrophobic stars to hydrophilic channels, where they coalesce and transport more efficiently.

#### 3.2.4. Comparison Between Three 3D Structural Arrays

The fog collection rates of bumpy structures, three-pointed star-shaped structures, and five-pointed star-shaped structures across four wettability samples are summarized in Figure 15, with surface areas detailed in Table 1.

The top flat surface dominated water collection for hydrophobic PP samples by providing droplet growth areas. In superhydrophilic PC samples and PC patterns on hydrophobic PP substrates, larger pattern areas allowed greater exposure to fog, enhancing collection rates. Hybrid wettability surfaces performed well across all structures, as the wettability gradient facilitated rapid water transport. The highest fog collection rates were 364.5 g/m^2^h for bumpy structures (superhydrophilic PC bumps on hydrophobic PP), 341.6 g/m^2^h for three-pointed stars, and 366.2 g/m^2^h for five-pointed stars (both hydrophobic PP stars on superhydrophilic PC substrates). Although star-shaped designs aimed to improve fog collection via Laplace pressure gradients, their performance was similar to bumpy structures, likely due to printing limitations reducing effectiveness at sharp tips. Enhancing 3D printing resolution could improve star-shaped designs and enable the exploration of alternative patterns for optimized fog harvesting.

## 4. Discussion

The hybrid wettability surfaces developed in this study demonstrated significantly improved fog collection efficiency compared to homogeneous surfaces, confirming the synergistic role of hydrophilic and hydrophobic regions. The bumpy structures achieved a maximum collection rate of 364.5 g/m^2^h, significantly outperforming homogeneous samples. This performance aligns with previous studies emphasizing the importance of hybrid wettability gradient for efficient droplet coalescence, and transport. The star-shaped structures achieved comparable results, with three- and five-pointed designs collecting at rates of 341.6 g/m^2^h and 366.2 g/m^2^h, respectively. However, the expected performance gains from these geometries were limited by printing resolution, particularly in forming sharp tips necessary to maximize Laplace pressure gradients. Advanced techniques such as lithography offer higher efficiency through precise nanoscale features, but the 3D printing and plasma treatment methods in this study provide a cost-effective alternative, despite current resolution limitations. Future efforts should improve 3D printing resolution and incorporate nanostructures to optimize efficiency. Integrating grooves with bumpy or star-shaped designs may further enhance performance by combining directional transport. Real-world testing is crucial to validate scalability for large-scale applications.

## 5. Conclusions

This study successfully optimized a two-step fabrication process combining FDM 3D printing and atmospheric pressure plasma treatment to create hybrid wettability surfaces inspired by the Namib Desert beetle and cacti. Our primary goal was to develop a more facile and eco-friendly approach for fabricating hybrid wettability surfaces, which was successfully achieved. The results demonstrated that hybrid surfaces with hydrophilic and hydrophobic regions significantly enhanced fog collection efficiency. Bumpy structures achieved the highest collection rate of 364.5 g/m^2^h, while star-shaped patterns performed comparably despite limitations in printing resolution. Our findings highlight that hybrid wettability surfaces consistently outperform homogeneous ones in fog collection due to wettability gradient and enhanced transport. This confirms the effectiveness of combining simple structural designs with hybrid wettability regions to enhance water transport and droplet coalescence. The study validates the practicality of a cost-effective and scalable fabrication method for developing bio-inspired surfaces, providing a foundation for further innovations in fog harvesting and related applications.

## Figures and Tables

**Figure 1 biomimetics-10-00143-f001:**
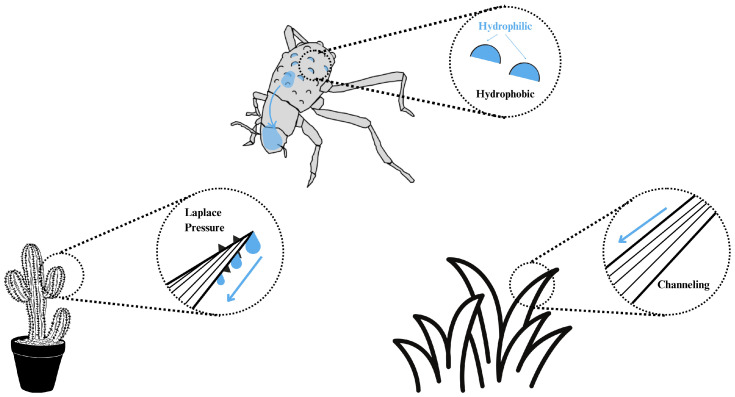
Summary of desert animals and plants harvesting water from fog without energy absorption. The blue arrow indicates the transport direction of collected droplets.

**Figure 2 biomimetics-10-00143-f002:**
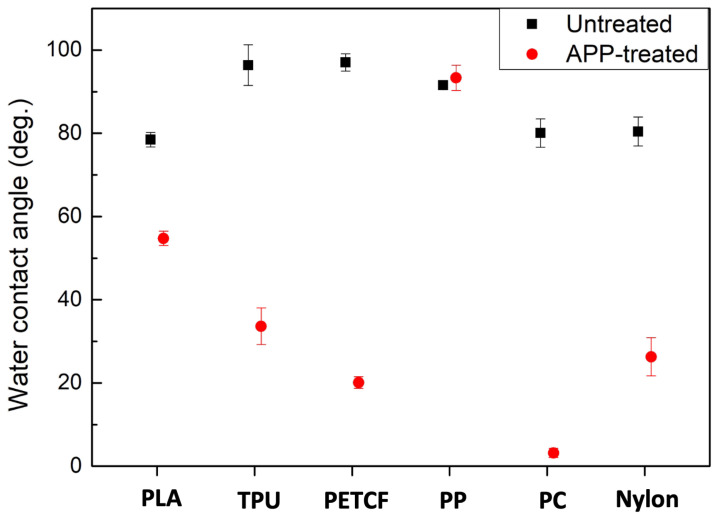
Water contact angles of six materials before and within one minute after AP plasma treatment. PP maintained its hydrophobicity with a contact angle of 93.3° after AP plasma treatment, while PC became super hydrophilic with a contact angle of 3.2°.

**Figure 3 biomimetics-10-00143-f003:**
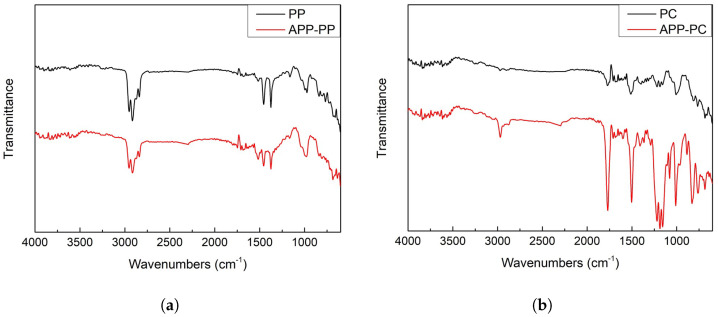
FTIR spectra of (**a**) untreated PP and APP-treated PP and (**b**) untreated PC and APP-treated PC samples.

**Figure 4 biomimetics-10-00143-f004:**
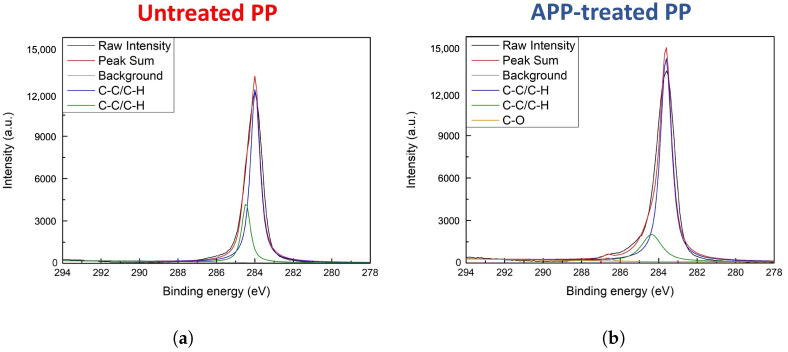
C1s ESCA spectra (**a**,**b**) of untreated PP and APP-treated PP. O1s ESCA spectra (**c**,**d**) of untreated PP and APP-treated PP.

**Figure 5 biomimetics-10-00143-f005:**
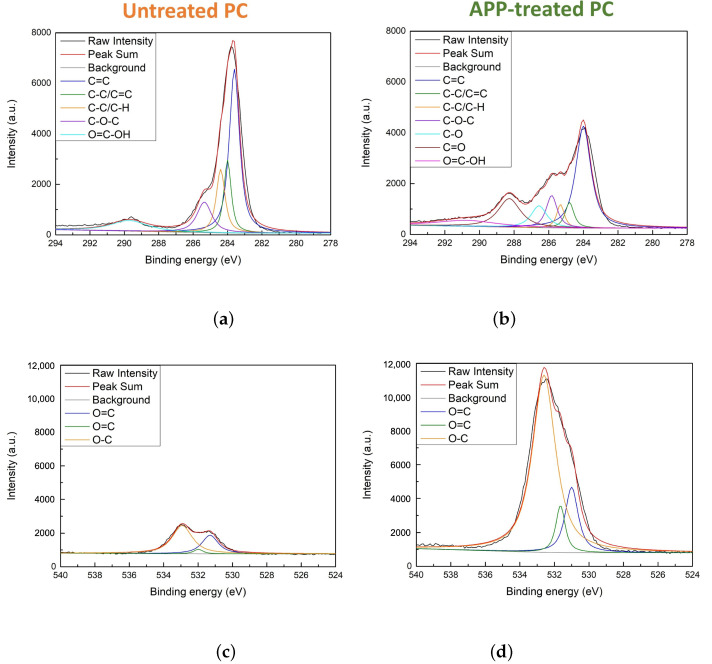
C1s ESCA spectra (**a**,**b**) of untreated PC and APP-treated PC. O1s ESCA spectra (**c**,**d**) of untreated PC and APP-treated PC.

**Figure 6 biomimetics-10-00143-f006:**
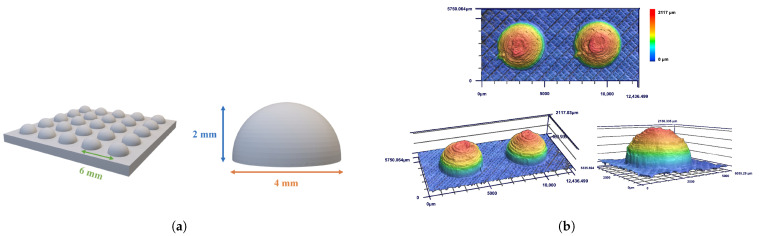
(**a**) The designed pattern of the beetle-inspired bumpy structure. The diameter of bumps is 4 mm, the center-to-center spacing of bumps is 6 mm, and the height of bumps is 2 mm. (**b**) Laser confocal images of beetle-inspired bumpy structures.

**Figure 7 biomimetics-10-00143-f007:**
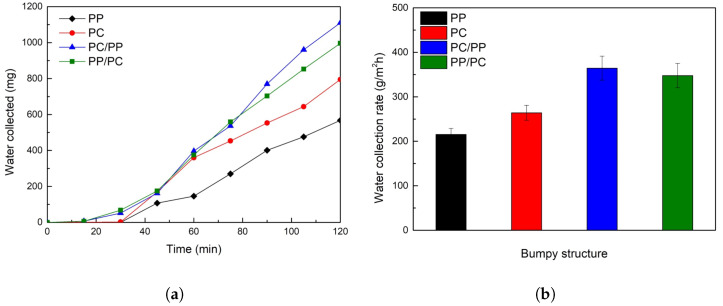
(**a**) Plot of collected water from bumpy structures over time for 120 min. (**b**) Water collection rates of bumpy structures. (PP: black, PC: red, PC/PP: blue, PP/PC: green).

**Figure 8 biomimetics-10-00143-f008:**
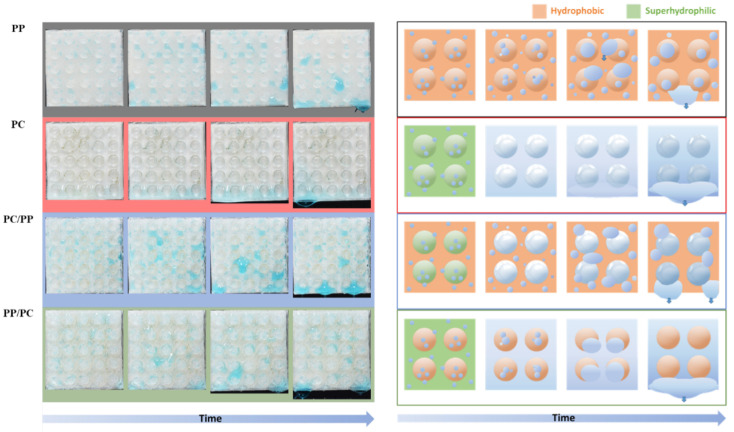
Photographs and illustrations of fog collection processes for bumpy structures. From top to the bottom are PP (black), PC (red), PC bumps on PP substrate (blue), and PP bumps on PC substrate (green).

**Figure 9 biomimetics-10-00143-f009:**
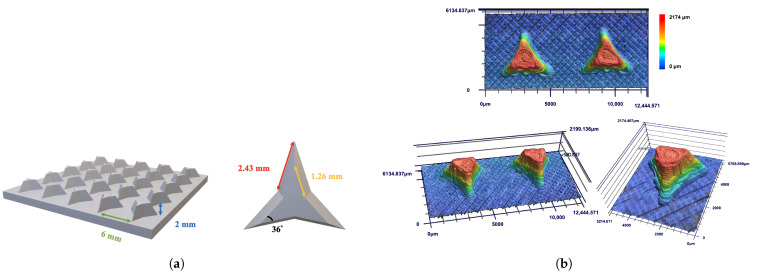
(**a**) The designed pattern of the three-pointed star-shaped structure. The top and bottom lengths of stars are 1.26 mm and 2.43 mm, the center-to-center spacing of stars is 6 mm, and the height of stars is 2 mm. (**b**) Laser confocal images of three-pointed star-shaped structures.

**Figure 10 biomimetics-10-00143-f010:**
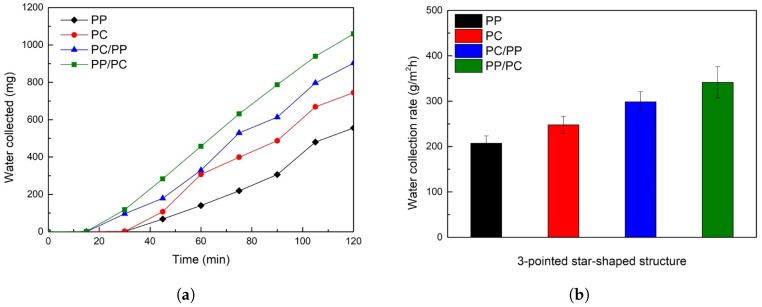
(**a**) Plot of collected water from three-pointed star-shaped structures over time for 120 min. (**b**) Water collection rates of three-pointed star-shaped structures (PP: black, PC: red, PC/PP: blue, PP/PC: green).

**Figure 11 biomimetics-10-00143-f011:**
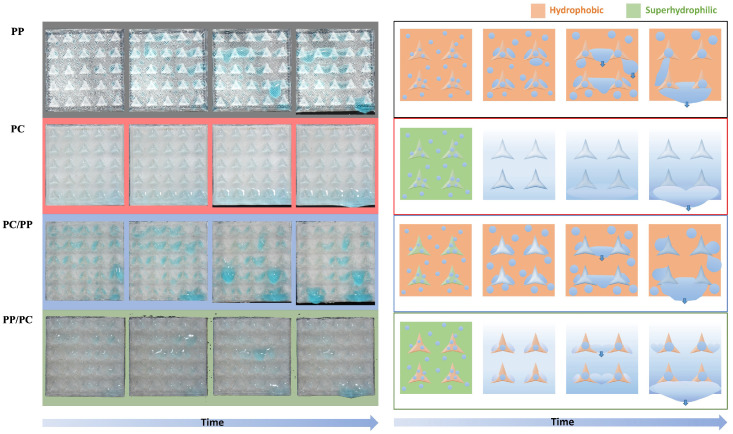
Photographs and illustrations of fog collection processes for three-pointed star-shaped structures. From top to the bottom are PP (black), PC (red), PC bumps on PP substrate (blue), and PP bumps on PC substrate (green).

**Figure 12 biomimetics-10-00143-f012:**
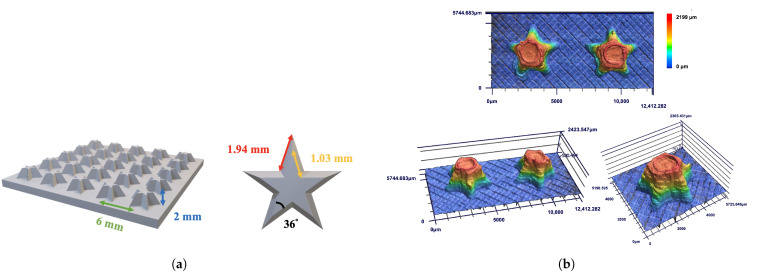
(**a**) The designed pattern of the five-pointed star-shaped structure. The top and bottom length of stars are 1.03 mm and 1.94 mm, the center-to-center spacing of stars is 6 mm, and the height of stars is 2 mm. (**b**) Laser confocal images of five-pointed star-shaped structures.

**Figure 13 biomimetics-10-00143-f013:**
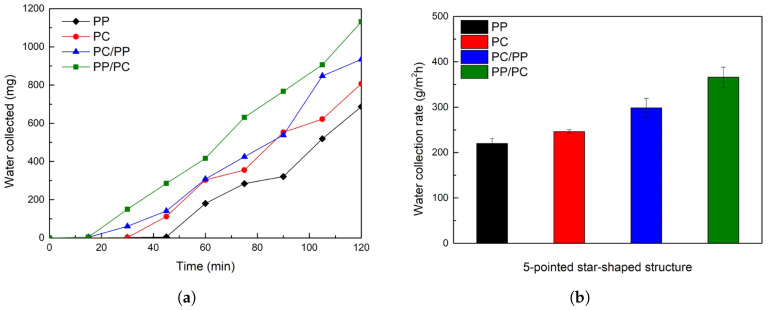
(**a**) Plot of collected water from five-pointed star-shaped structures over time for 120 min. (**b**) Water collection rates of three-pointed star-shaped structures (PP: black, PC: red, PC/PP: blue, PP/PC: green).

**Figure 14 biomimetics-10-00143-f014:**
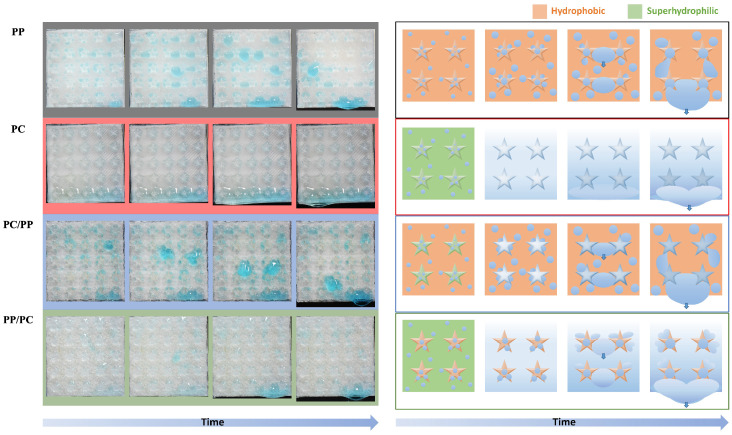
Photographs and illustrations of fog collection processes for five-pointed star-shaped structures. From top to the bottom are PP (black), PC (red), PC bumps on PP substrate (blue), and PP bumps on PC substrate (green).

**Figure 15 biomimetics-10-00143-f015:**
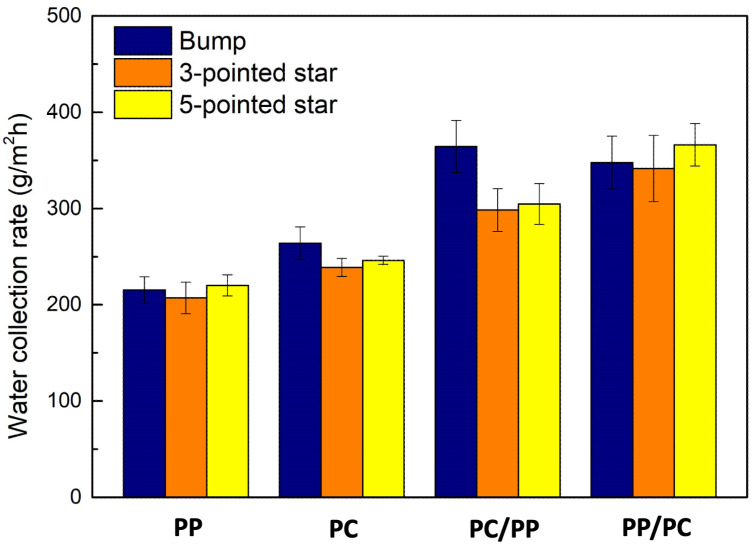
Water collection rates of bumpy, three-pointed, and five-pointed, star-shaped structures.

**Table 1 biomimetics-10-00143-t001:** Surface area of bumpy structures, 3-pointed star-shaped structures, and 5-pointed star-shaped structures.

Structure	Top Flat Area	Pattern Area	Substrate Area	Total Area
	(mm^2^)	(mm^2^)	(mm^2^)	(mm^2^)
Bump	-	674.45	729.89	1404.34
3-pointed star	41.65	562.75	833.62	1396.37
5-pointed star	56.15	625.08	820.75	1445.83

## Data Availability

Data are contained within the article and Appendix A.

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
