# Peer review of "Desert Beetle-Inspired Hybrid Wettability Surfaces for Fog Collection Fabricated by 3D Printing and Atmospheric Pressure Plasma Treatment"

_biomimetics, 2025, doi:10.3390/biomimetics10030143_

Round 1

Reviewer 1 Report

Comments and Suggestions for Authors

his study presents a cost-effective method for fabricating patterned surfaces with hybrid wetting properties and 3D structuring. While the fabrication approach appears novel, the explanation lacks sufficient detail, and the significance of the results remains unclear. Below are specific comments regarding different sections of the manuscript.

Introduction

The concept of biphilic surfaces for fog collection is well-established, with numerous studies demonstrating similar approaches in two-dimensional (2D) configurations. Although this work focuses on transitioning to three-dimensional (3D) patterns, it is essential to acknowledge prior studies that serve as foundational references. The following works should be cited:

  • Garrod et al., Langmuir, 2007, 23(2), 689–693. https://doi.org/10.1021/la0610856

  • Bai et al., Adv. Mater., 2014, 26(29), 5025–5030. https://doi.org/10.1002/adma.201400262

  • Nioras et al., ACS Appl. Nano Mater., 2022, 5(8), 11334–11341. https://doi.org/10.1021/acsanm.2c02439

Materials and Methods

The fabrication method for hybrid surfaces requires more explanation. Specifically:

  • Can the material be changed mid-printing? If so, how is this achieved?

  • What exact steps are involved in fabricating the hybrid surfaces? A clearer, step-by-step description is necessary.

Surface Characterization

A key observation that needs further clarification is why polypropylene (PP) remains hydrophobic after atmospheric-pressure plasma (APP) treatment, whereas other materials with similar or higher static contact angles (SCAs) become hydrophilic or superhydrophilic. This phenomenon should be explained, preferably with references to prior studies that discuss the plasma treatment of PP or similar materials.

Fog Collection Efficiency

Several points in this section require clarification:

  • The meaning of the acronyms PP/PC and PC/PP should be explicitly stated upon first use. Which component forms the bumps, and which serves as the background?

  • The term “condensation” is incorrectly used in the context of fog collection. Condensation refers to a gas-to-liquid phase transition, whereas fog collection involves the accumulation and coalescence of pre-existing liquid droplets. A more accurate term should be used.

  • The performance of hydrophobic bumps on a superhydrophilic background is surprising, given that uniform superhydrophilic surfaces do not perform well. The authors should provide a clear explanation for this observation.

  • The reported fog collection rate appears relatively low compared to similar studies, many of which report values in the range of several grams/cm²/h. The manuscript should discuss possible reasons for this discrepancy and consider introducing alternative performance metrics. The following work may provide useful insights:

    • Nioras et al., ACS Appl. Mater. Interfaces, 2021, 13(40), 48322–48332. https://doi.org/10.1021/acsami.1c16609

Conclusion

Overall, while the proposed fabrication method is promising, the manuscript requires significant improvements in clarity, explanation, and contextualization within existing literature. Further elaboration on fabrication techniques, material behavior under APP treatment, and the significance of results compared to prior work would strengthen the study’s impact.

Author Response

Please see the attachment in the box. 

Reviewer 2 Report

Comments and Suggestions for Authors

Dear Authors,

This manuscript investigated the biomimetic hybrid wettability surfaces with varied topographical features inspired by the Desert Beetle for fog harvesting, using FDM multi-material (PP and PC) 3D printing and Atmospheric Pressure Plasma Treatment. The study is relevant to the field of Biomimetics. The proposed method is cost-effective, and the experimental setup is well-designed. However, there are some concerns regarding the data analysis and discussion, which I think should be addressed before the manuscript is accepted for publication.

Here are my comments:

#1: The authors treated various polymer surfaces using Atmospheric Pressure Plasma Treatment and measured the changes in Water Contact Angle (WCA) before and after treatment, as presented in Figure 2. However, it is known that the wettability of plasma-treated polymer surfaces changes over time due to surface aging or hydrophobic recovery. The authors should specify how long after plasma treatment the WCA measurements were conducted. Additionally, it would be beneficial to provide the temporal evolution of WCA for PP and PC surfaces after plasma treatment to illustrate the extent of hydrophobic recovery.

#2: When analyzing the fog-harvesting performance of different surface structures, the authors presented Water Collection Rate (Fig. 7b, Fig. 10b, Fig. 13b, and Fig. 15). Since the fog-harvesting rate is not a constant value but fluctuates over time, the authors should clarify over what time period the reported collection rates were averaged to ensure consistency and reproducibility of the results.

#3: The manuscript presented three distinct topographical structures: Bumpy Structures, Three-Pointed Star-Shaped Structures, and Five-Pointed Star-Shaped Structures. Within each structure type, four different surface compositions were tested: PP, PC, PC/PP, and PP/PC. The authors attributed the superior performance of PC/PP and PP/PC surfaces over homogeneous PP and PC surfaces in fog-harvesting to two key factors: Water droplet condensation and Water transportation. However, the authors did not explicitly explain why homogeneous PP and PC surfaces performed worse in these aspects compared to hybrid wettability surfaces. I suggest that in the Introduction (at the end of the first paragraph of the Introduction), the authors propose a hypothesis, based on the water collection mechanisms of the Desert Beetle and other desert biological surfaces in fog-harvesting, that the hybrid wettability and topographical structures plays a role in fog-harvesting. Additionally, the authors should briefly explain why homogeneous surfaces are less effective than hybrid wettability surfaces in fog-harvesting even they have same topographical structures. This would help readers better understand the novelty of the work and provide a stronger theoretical foundation for the experimental results.

Thank you very much.

Yours sincerely,

Xxx

Author Response

Please see the attachment in the box.

Round 2

Reviewer 1 Report

Comments and Suggestions for Authors

The authors have carefully answered to all my comments and the manuscript can now be accepted as is.